# Real-World Use of Dalbavancin for Treatment of Soft Tissue and Bone Infection in Children: Safe, Effective and Hospital-Time Sparing

**DOI:** 10.3390/children11010078

**Published:** 2024-01-09

**Authors:** Désirée Caselli, Marcello Mariani, Claudia Colomba, Chiara Ferrecchi, Claudio Cafagno, Daniela Trotta, Ines Carloni, Daniela Dibello, Elio Castagnola, Maurizio Aricò

**Affiliations:** 1Infectious Diseases, Children’s Hospital Giovanni XXIII, Azienda Ospedaliero Universitaria Consorziale Policlinico di Bari, 70124 Bari, Italy; desiree.caselli@policlinico.ba.it (D.C.); claudio.cafagno@policlinico.ba.it (C.C.); 2Pediatrics and Infectious Diseases Unit, IRCCS Istituto Giannina Gaslini, 16147 Genoa, Italy; 3Department of Health Promotion, Maternal and Child Care, Internal Medicine and Medical Specialties “G. D’Alessandro”, University of Palermo, 90133 Palermo, Italy; claudia.colomba@libero.it; 4Division of Pediatric Infectious Diseases, “G. Di Cristina” Hospital, ARNAS Civico Di Cristina Benfratelli, 90127 Palermo, Italy; 5Department of Neurosciences, Rehabilitation, Ophthalmology, Genetics and Maternal and Child Sciences (DINOGMI), University of Genoa, 16126 Genoa, Italy; s4034483@studenti.unige.it; 6Pediatrics, S. Spirito Hospital, A.S.L. Pescara, 65124 Pescara, Italy; daniela.trotta@asl.pe.it (D.T.); maurizio.arico@asl.pe.it (M.A.); 7Department of Mother and Child Health, Salesi Children’s Hospital, 60123 Ancona, Italy; ines.carloni@ospedaliriuniti.marche.it; 8Pediatric Orthopedics and Traumatology, Children’s Hospital Giovanni XXIII, 70125 Bari, Italy; daniela.dibello@policlinico.ba.it

**Keywords:** dalbavancin, soft tissue infection, acute bacterial skin and skin structure infections (ABSSSIs), bone infection

## Abstract

Acute bacterial skin and skin structure infections (ABSSSI) and osteoarticular infections compound the burden of morbidity, mortality and prolonged hospitalizations among gram-positive infections. Dalbavancin, a second-generation, intravenous lipoglycopeptide, due to its prolonged half-life, can be a valuable alternative in their treatment when administered as inpatient treatment at the price of an extended hospital stay. Between October 2019 and September 2023, 31 children and adolescents were treated with dalbavancin because of bone and joint infections (n = 12 patients, 39%), ABSSSI (n = 13 patients, 42%), mainly for the limbs, facial cellulitis or complicated ABSSSI (n = 6 patients, 19%), at five Italian pediatric centers. Microbiological study provided gram-positive bacterial isolate in 16 cases, in 11 cases from a positive blood culture; 9 of them were MRSA. Twenty-five patients were initially treated with a different antibiotic therapy: beta-lactam-based in 18 patients (58%), glycopeptide-based in 15 patients (48%) and daptomycin in 6 (19%). The median time that elapsed between admission and start of dalbavancin was 18 days. A total of 61 doses of dalbavancin were administered to the 31 patients: 16 received a single dose while the remaining 15 patients received between two (n = 9) and nine doses. The frequency of administration was weekly in five cases or fortnightly in nine patients. Median length of stay in hospital was 16 days. Median time to discharge after the first dose of dalbavancin was 1 day. Treatment was very well-tolerated: of the 61 administered doses, only four doses, administered to four patients, were associated with an adverse event: drug extravasation during intravenous administration occurred in two patients, with no sequelae; however, in two patients the first administration was stopped soon after infusion start: in one (ID #11), due to headache and vomiting; in another (ID #12) due to a systemic reaction. In both patients, drug infusion was not repeated. None of the remaining 29 patients reported treatment failure (resistant or recurrent disease) or an adverse effect during a median follow-up time of two months. The use of dalbavancin was safe, feasible and also effective in shortening the hospital stay in children and adolescents.

## 1. Introduction

Gram-positive bacteria represent a frequent, well-known cause of infection in Europe, with an estimated expense for European healthcare systems of €380 million only for methicillin-resistant *Staphylococcus aureus* (MRSA) infections, which affect about 150,000 people per year [1]. Acute bacterial skin and skin structure infections (ABSSSI) are among the most common infections requiring hospitalization and exert a substantial burden on the health care system. This term comprises the following clinical pictures: cellulitis/erysipelas, a diffuse skin infection characterized by spreading areas of redness, edema and/or induration; wound infection, characterized by purulent drainage from a wound with surrounding redness, edema and/or induration; major cutaneous abscess, characterized by a collection of pus within the dermis or deeper that is accompanied by redness, edema and/or induration.

ABSSSI result in approximately 3 million pediatric health care visits per year in the United States [2]. Over the last 15 years, increasing occurrence of community-associated methicillin-resistant *staphylococcus aureus* caused the number of hospitalizations for ABSSSI in children to more than double [3,4], now exceeding over 70,000 per year [4]. Such hospitalizations have been associated with an increasing cost of care and need for invasive procedures [4].

Although ABSSSI are the most common type of infection caused by gram-positive agents, the major burden in terms of morbidity, mortality, prolonged hospitalizations and costs is sustained by gram-positive-related cardiovascular and osteoarticular infections [5]. This concept applies also to pediatric patients. Although timely diagnosis and appropriate treatment results in a cure, in some cases it may require extended hospitalization and life-threatening complications [6,7]. Since the year 2000, rates of hospitalizations among pediatric patients with skin and skin structure infections have increased rapidly. A 2013 survey showed doubling incidence between 1997 and 2009 [8].

In Italy about 30% of *Staphylococcus aureus* are MRSA [9] thus requiring a specific treatment. Vancomycin is usually considered the first-choice agent due to its efficacy and limited toxicity. However, intravenous (i.v.) infusion requires an extended hospital stay, especially for treatment of bone infection, and achievement of therapeutic pharmacokinetic/pharmacodynamic (PK/PD) targets could be difficult [9]. Thus, availability of a long-acting agent appears appealing.

Dalbavancin is a second-generation, intravenous lipoglycopeptide. Due to its favorable PKs characterized by a terminal half-life of 2 weeks—related in part to 93% protein binding—high bactericidal activity against most gram-positive bacteria, good safety profile and high tissue penetration, dalbavancin can be a valuable alternative in the treatment of ABSSSI as well as bone and joint infections [10,11,12,13,14]. Recommended use of dalbavancin is intravenous infusion over 30 min, at a dose of 1500 mg for adults, given either as a single infusion or as 1000 mg in the first week followed by 500 mg in the following week. On 13 October 2022, the Committee for Medicinal Products for Human Use (CHMP) adopted a positive opinion recommending a change to its terms of marketing authorization, including the treatment of pediatric patients from the age of 3 months [15]. For children, the dose depends on age and body weight and should be no more than 1500 mg [15].

The aim of this retrospective analysis is to collect and report the experience of five Italian pediatric centers in the use of dalbavancin for off-label treatment of bone infections and in-label ABSSSI in children and adolescents.

## 2. Patients and Methods

In this retrospective observational study, we reviewed the clinical records of all children and adolescents with definite or highly suspected diagnoses of gram-positive bacterial skin and soft tissue or bone and joint infections, treated with dalbavancin at the five participating centers: Pediatric Infectious Diseases, Children’s Hospital, Bari; Pediatrics and Infectious Diseases, Genoa; Pediatric Infectious Diseases, Children Hospital, Palermo; Pediatrics, Pescara; Pediatrics, Ancona. 

Inclusion criteria were age between 3 months and less than 19 years and ABSSSI or bone infection requiring i.v. treatment. Exclusion criteria were treatment refusal or an unavailable clinical record. 

Based on the site of infection, enrolled patients were divided into three groups [16]:

ABSSSI, mainly limbs or facial cellulitis; complicated ABSSSI: patients who presented with an ABSSSI but in whom clinical complications were subsequently diagnosed, such as bone or joint involvement or other abscess; bone and joint infection, including long-bone osteomyelitis, prosthetic joint infection, septic arthritis or native vertebral infection.

The following data were collected from patient medical charts: age and gender; type and site of infection; associated condition; bacterial isolate; antibiotic therapy preceding the use of dalbavancin; duration of hospital stay; dose and number of administered doses of dalbavancin; adverse effects; patient outcome at the time of writing or at last follow-up. Anonymized data were collected in a specific access database (Microsoft™ Office™ Professional plus 2016, Microsoft Corporation, Redmond, WA, USA).

Informed consent for treatment and for data analysis for scientific purposes was obtained from the parents or legal guardians for all patients. The study was conducted in accordance with the Declaration of Helsinki Ethical Principles and Good Clinical Practices. Institutional Review Board approval was waived due to study design.

## 3. Results

Between October 2019 and September 2023, 31 children and adolescents were treated with dalbavancin because of osteomyelitis or ABSSSI at five Italian pediatric centers (Genova, n = 14; Pescara, n = 7; Bari, n = 5; Palermo, n = 4; Ancona, n = 1). Based on the site of infection, enrolled patients were divided as follows: ABSSSI (n = 13 patients, 42%), mainly limbs or facial cellulitis; complicated ABSSSI (n = 6 patients, 19%), patients who presented with an ABSSSI but in whom clinical complications were subsequently diagnosed, such as bone or joint involvement or other abscess; bone and joint infections (n = 12 patients, 39%), including native vertebral infections, long-bone osteomyelitis, prosthetic joint infection and septic arthritis.

Their demographic and presenting features are summarized in Table 1. Overall, 20/31 (64%) are male, with a median age of 12.3 years (range, <1 to 18 years). A clinically relevant associated or potentially predisposing condition/event was reported in 11 patients.

Microbiological study provided gram-positive bacterial isolate in 16 cases, in 11 cases from positive blood culture; 9 of them were MRSA. 

In patients with bone infection, the lower limb was involved in eight patients and the skull in three. 

Twenty-five patients were initially treated with another antibiotic therapy, which was beta-lactam-based in 10 patients (40%), glycopeptide-based in 14 patients (56%) and daptomycin in one. 

The administered dose of dalbavancin was of 18 mg/kg in 20 patients and 22.5 mg/kg in 8 patients while in two patients it was lower than 18 and in two patients the maximum dose of 1500 mg was given. 

A total of 61 doses of dalbavancin were administered to the 31 patients: 16 received a single dose while the remaining 15 patients received between two (n = 9) and nine doses. The frequency of administration was weekly in five cases or fortnightly in nine patients while in one case three doses were given at weeks 1, 2 and 4. Details of treatment are shown in Table 1. 

Length of hospital stay ranged between 2 and 150 days (median, 16 days). Dalbavancin was started after a median of 18 days of hospital stay (range, 2 to 122). Time to discharge of patients after the first dose of dalbavancin in 20 patients ranged between <1 and 3 days (median, 1 day); in six patients dalbavancin was started after patient discharge, during outpatient care. 

Treatment was very well-tolerated: of the 61 administered doses, only in four doses, administered to four patients, was an adverse event reported: drug extravasation during intravenous administration occurred in two patients, with no sequelae. However, in two patients the first drug administration was stopped soon after infusion start: in one (ID #11) due to headache and vomiting; in another (ID #12) due to systemic reaction (urticarial rash, pruritus and flushing occurring within one hour from infusion start). In both patients, drug infusion was not repeated. 

It may be worth reporting that in one patient (case #18) with extended infection of the tool of osteosynthesis, dalbavancin therapy was effective in saving the tool from risk of removal. 

We did not observe any treatment failure of dalbavancin therapy. None of the 29 treated patients reported need for alternative/additional antibiotic therapy, nor any disease recurrence or any adverse effects during a follow-up time ranging between 1 and 14 months (median, 2 months). 

## 4. Discussion 

Bacterial infection by gram-positives is quite common in children and adolescents. The frequent involvement of soft tissues and bones may carry a heavy burden of morbidity and hospital stay, not excluding the risk of mortality. All the above represent a huge economic and assistance burden for the public health care system in our country.

Even non-complicated gram-positive infections with a low mortality rate (such as ABSSSI) represent a relevant concern due to their high prevalence in community settings along with the high percentage of recurrences and hospital re-admissions involved [17,18].

In this study, we describe the experience of treating ABSSSI and bone infection in a quite large group of 31 children or adolescents, with definite or highly-suspected diagnosis of gram-positive bacterial infection; most of them had been initially treated with daily intravenous therapy with standard antibiotics registered in Italy for hospital use only, which was subsequently switched to a dalbavancin regimen. Given the retrospective, real-life approach of the study, antimicrobial treatment schedules, including the switch to dalbavancin therapy and its duration, were prescribed exclusively based on the clinical judgment of the attending pediatric infectious diseases specialist, taking into consideration the site and severity of the infection and the isolated pathogens, available in one half of the cases.

In our experience, dalbavancin demonstrated great efficacy and tolerability, allowing a high rate of prompt clinical response, with a very low frequency of adverse events. It should be noted that rapid intravenous infusions of dalbavancin (<30 min) might cause reactions that resemble vancomycin infusion reaction (formerly “red man syndrome”) (e.g., flushing of the upper body, urticaria, pruritus, rash). Stopping or slowing the infusion may result in cessation of these reactions [19]; so the recommended 1 h infusion time should be strictly maintained. In a recent phase-3 international study, treatment-emergent adverse events occurred in 7.2% and 9.0% of patients in dalbavancin single-dose and dalbavancin 2-dose arms, respectively. Three serious adverse events occurred in the dalbavancin single dose arm, but no treatment-related adverse events leading to study discontinuation. The authors conclude that the safety profile of dalbavancin was consistent in children and adults with ABSSSI [13].

However, the most attractive feature of our experience of treatment with dalbavancin is the feasibility of administering it on an outpatient basis, or in an inpatient immediately before discharge. In this study, dalbavancin was not the initial choice for parenteral antibiotic therapy for all but three patients. This was partly due to its off-label use before extension of pediatric registration. Most patients had initial therapy with a beta-lactam agent and then shifted to a glycopeptide agent (usually vancomycin), also due to evidence of isolation of a gram-positive microorganism. The choice to introduce dalbavancin was usually driven by the wish of completing antibiotic therapy started by vancomycin (adopted in 80% of the patients) while shortening the hospital stay. All the above may explain the 18-day median duration of the hospital stay before administration of dalbavancin. 

However, the social cost of the extended number of hospital days is immediately clear to pediatricians and to families but cannot be overlooked by clinical governance when translating it into a higher cost of medical care. This use of the long-acting antimicrobial agent dalbavancin was associated with significant savings in the many antimicrobial treatment-related costs, the first of which is bed occupancy. In a recent study, aimed at evaluating the impact of dalbavancin therapy on both hospital length-of-stay and treatment-related costs, in a cohort of 50 patients with diverse gram-positive bacterial infections (ABSSSI, 12 patients; ABSSS-C, 8 patients; osteoarticular infections, 18 patients; vascular graft or cardiovascular implantable electronic device infections, 12 patients), by switching to dalbavancin, a median of €8259 and 14 hospital days per patient were saved [16].

Treatment schedule turned out to be quite variable, in keeping with the real-life, retrospective analysis of this series. Of the 31 patients, one half received a single dose while the remaining half received a number of doses ranging between 2 and 9, with a weekly frequency of administration in five cases and fortnightly in nine patients. In their large series of adults treated with dalbavancin, Poliseno et al., describe a subgroup of patients receiving additional dose(s) of dalbavancin one or more weeks apart [16]. Consolidation of the treatment with a lower dose of dalbavancin (e.g., 1000 mg every 14 days following the first dose of 1500 mg) could be used in these complex infections [20,21,22,23,24,25], and simply two weekly 1500 mg doses of dalbavancin have been proposed as an effective treatment of osteomyelitis and complicated bacteremia or endocarditis due to gram-positive bacteria [26]. In our experience, in a single patient with bone and soft tissue infections (case #26), we ended up administering 9 weekly doses of dalbavancin to be able to achieve complete clinical and imaging control of an extended infection involving the iliac bone, femur and surrounding soft tissues. This was the only case of extended duration of this therapy in our group.

However, the most attractive feature of this strategy is definitely the opportunity to more quickly discharge a patient who has been (or might alternatively be) admitted as an inpatient for weeks for parenteral anti-staphylococcal therapy. This is attractive not only in adults but also in children, given the social burden of having a child and a parent staying in the hospital for quite a long time and the inherent burden of psychological issues, risk of intra-hospital infections and days lost at school for the patient and at work for the parents. The “early discharge” strategy built on dalbavancin as a long-acting antimicrobial turned out to be effective and time saving for the pediatric patient and his/her family. In our experience, it provides a safe and very effective treatment for bone infections and ABSSSI in children and adolescents, also allowing the hospital to reduce the length of stay of the child, thus offering the spared bed-days to other patients; a cost-saving and socially favorable approach. This is well-reflected in the comparison of the extended duration of the hospital stay before administration of dalbavancin (median, 18 days) with the number of days between dalbavancin administration and discharge, which in 20 patients ranged between <1 and 3 days (median, 1 day), with six additional patients receiving dalbavancin first dose after discharge, as outpatient care.

Finally, even a child or adolescent treated for an extended duration, i.e., repeated administration of long-lasting antibiotics, would not require daily venipuncture with possible implanting of central venous catheters and would dramatically reduce the number of visits to the hospital, which is favorable for the hospital team and organization but also for the lowered risk of hospital-acquired (additional) infection.

In 2017, the Italian Society of Infectious Diseases stated in a consensus paper [27] that a new drug, dalbavancin, could have been suitable for early discharge, thus changing the management of ABSSSI with a considerable reduction of hospitalization costs and related risks. Our experience widely confirms that statement. 

In this study, the participating centers are scattered all over the country (one in the north, one in the center and three in the south) and range from large, pediatric referral hospitals to smaller pediatric wards embedded in a large general hospital. Thus, we do not see limitations deriving from geographic or setting-type as reasons against wider use of dalbavancin in such patients.

Our study has limitations: although all patients treated in the participating institutions were reported, it is possible that some potentially eligible patients did not receive this treatment. Although collected from five different institutions, the number of patients is relatively small, and reproduction on a larger cohort might be warranted. The study population is highly variable in age, diagnosis and treatment schedule. Furthermore, since this is a retrospective analysis and not a prospective trial, the criteria for shifting from glycopeptide or beta-lactam antibiotic to dalbavancin could be better defined for a prospective study.

In conclusion, recent extension of the use of dalbavancin to younger patients extends the treatment options for children and adolescents with ABSSSI and bone and joint infection. Although conventional therapy with glycopeptides, especially vancomycin, offers an excellent success rate in the vast majority of patients, extended and repeated i.v. infusion of the antibiotic during the day requires an extended hospital stay for the child and his/her parent. Beyond the obvious discomfort for the patient and the entire family, this raises the issue of the cost of care and the related use of hospital beds, which can be inappropriate. Thus, prospective studies focusing on criteria for the use of dalbavancin as a first-line agent in children and adolescents with ABSSSI and bone and joint infection appears warranted. In a prospective trial, number of doses and duration of therapy could be uniformly defined, thus allowing for the design of an optimal treatment schedule in children with this engaging but quite common infection.

## Figures and Tables

**Table 1 children-11-00078-t001:** Presenting features, pathogen identification, previous antibiotic therapy, detail of dalbavancin therapy, hospital days and treatment outcome in 31 children and adolescents with acute bacterial skin and skin structure infections (ABSSSI), complicated ABSSSI or bone infection.

Group	Patient			Infection		Previous Therapy		Dalbavancin Therapy		Hospital Days		Outcome, Follow-Up Time
	ID	Gender Age (y)	Associated Condition	Site	Bacterial Isolate	Antibiotics	Days	mg/kg	Doses (week)	Total	From Dalbavancin to Discharge	
ABSSSI	5	F/8	No	Subcutaneous abscess, thigh	MRSA (abscess drainage)	Beta-lactam + Daptomycin	22	18	2 (w 1,3)	22	0	well at 1 m.
ABSSSI	34	F/1	no	abscess, thigh injection site		Daptomycin	10	22.5	1	11	1	well at 6 m.
ABSSSI	33	F/13	Poikiloderma with neutropenia	panniculitis		Carbapenem, Daptomycin	7	18	1	8	1	well at 6 m.
ABSSSI	32	F/7	No	Cellulitis, orbit		Beta-lactam + Daptomycin	122	18	1	123	1	well at 6 m.
ABSSSI	14	M/1	No	Cellulitis, thigh	MSSA (swab)	Vancomycin	9	22	1	9	0	well at 1 m.
ABSSSI	2	M/16	No	Cellulitis and fasciitis, leg		Beta-lactam, Teicoplanin, Clindamycin	22	1500 mg	2 (w 1,3)	22	0	extravasation; well at 1 m.
ABSSSI	3	M/15	No	Lymphadenitis, cervical		Vancomycin + Beta-lactam	27	18	2 (w 1,3)	27	0	well at 1 m.
ABSSSI	4	M/0.8	No	Cervical abscess	*S. pyogenes * (blood)	Vancomycin + Beta-lactam	31	22.5	1	32	1	well at 1 m.
ABSSSI	15	M/16	No	Cellulitis, abdominal wall		Beta-lactam	2	18	1	5	3	well at 3 m.
ABSSSI	13	M/15	No	Cellulitis, inguinal		Beta-lactam	9	18	1	9	0	well at 3 m.
ABSSSI	12	F/14	No	Cellulitis, submental		Vancomycin	6	11	1	9	3	Systemic reaction, stop infusion, shift to other treatment
ABSSSI	11	F/11	IgA vasculitis	Cellulitis, foot		Beta-lactam	20	15	1	3	PD	Headache and vomiting: stop infusion, shift to other treatment
ABSSSI	10	M/9	Autism	Skin cellulitis, recurrent	MRSA (swab)	-	-	18	1	2	0	well at 1 m.
ABSSSI-C	1	F/0.4	No	Abscess, subperiosteal femur	MSSA (blood)	Vancomycin + Beta-lactam	31	22.5	1	32	1	Femur fracture; well at 3 m.
ABSSSI-C	9	M/1	No	Pneumonia, pleural empyema	MRSA (pleural drainage)	-	-	22.5	1	29	2	well at 1 m.
ABSSSI-C	8	F/2	Intestinal pseudo-obstruction, chronic	Septic thrombosis	MRSA (blood)	Vancomycin	13	22.5	2 (w 1,3)	16	3	well at 1 m.
ABSSSI-C	6	F/18	Glomerulosclerosis, focal	Soft tissues and septicemia	MSSA (blood)	Vancomycin; Beta-lactam	12	22.5	1	26	14	Respiratory distress after infusion. Well at 2 m.
ABSSSI-C	31	M/12	no	Cellulitis, knee;	*Str. pyogenes* (blood)	Teicoplanin + Clindamycin	22	18	1	24	2	well at 7 m.
ABSSSI-C	7	M/14	No	Skin and soft tissue	MRSA (swab, skin lesion)	Beta-lactam, Vancomycin	20	18	1	6	1	well at 2 m.
BONE	17	M/12	No	Frontoparietal-sphenoidal epidural empyema	MSSA *S. intermedius* (abscess drainage)	Beta-lactam, Vancomycin	18	18	3 (w 1–3)	19	1	extravasation well at 2 m.
BONE	35	M/10	no	Femur	*S. aureus* PVL+, swab	Many	55	18	1	56	1	well at 3 m.
BONE	18	M/13	Foot surgery	Foot, following orthopedic surgery	MSSA, skin swab	Vancomycin + Beta-lactam	10	18	6 (w 1–6)	10	0	well at 12 m.
BONE	19	M/17	Lower limb asymmetry	Femur, following surgery	*S. hominis* & epidermidis (blood)	Beta-lactam	14	1500 mg	2 (w 1,3)	14	0	well at 2 m.
BONE	20	F/17	No	Mandible	*S. parasanguinis* (blood)	Beta-lactam; Clindamycin	77	18	3 (w 1–3)	0	PD	well at 11 m.
BONE	21	F/10	SBIDDS, citrullinemia	Femur	-	Beta-lactam; Teicoplanin	16	18	2 (w 1,3)	15	NR	well at 2 m.
BONE	22	M/13	Epilepsy	Surgical wound infection	MRSA (wound swab)	Daptomycin	105	18	2 (w 1,3)	36	PD	well at 2 m.
BONE	23	M/12	No	Foot	MRSA (blood)	Vancomycin; Daptomycin, Rifampicin	17	18	2 (w 1,3)	17	PD	well at 2 m.
BONE	24	M/9	No	Sacral vertebra	-	-	-	18	3 (w 1,2,4)	6	0	well at 2 m.
BONE	25	M/10	No	Tibia, pyomyositis	MRSA (blood)	Linezolid	53	18	2 (w 1,3)	12	PD	well at 14 m.
BONE	26	M/15	Primary immune deficiency	Femur and iliac bone; pyomyositis, gluteal; bilateral pneumonia, septicemia	MSSA (blood)	Vancomycin, Beta-lactam	40	17	9 (weekly)	41	1	well at 10 m.
BONE	16	M/1	Acute myeloid leukemia	Iliac bone	MRSA (blood)	Beta-lactam, Linezolid; others	90	22.5	3 (w 1–3)	150	PD	well at 1 m. post stem cell transplant

ABSSSI-C: complicated ABSSSI; SBIDDS: Short Stature, Brachydactyly, Intellectual Developmental Disability and Seizures syndrome; MRSA: Methicillin-Resistant *Staphylococcus aureus*; MSSA: Methicillin-Sensitive *Staphylococcus aureus*; PVL: Panton–Valentine leukocidin; PD: Post-discharge.

## Data Availability

The data presented in this study are available in article.

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
