# Peer review of "Real-World Use of Dalbavancin for Treatment of Soft Tissue and Bone Infection in Children: Safe, Effective and Hospital-Time Sparing"

_children, 2024, doi:10.3390/children11010078_

Round 1

Reviewer 1 Report

Comments and Suggestions for Authors

Many thanks for the opportunity to review this manuscript. The authors describe their experience of using dalbavancin in a paediatric population in Italy for skin and skin structure and orthopaedic infections.

I should commend the authors for producing the manuscript, but there are a few minor issues to address. I’d consider revising the title – perhaps “the real-world experience of…”

Line 24: Revise “as in-patient”

Line 29: Median

Line 30  and through-out the manuscript – beta-lactam

Line 30-31: The numbers in the abstract seem to be different to the data provided in the tables.

Line 32: is it 51 or 61 doses?

Line 48: Conventionally written as met(h)icillin

Line 62: First use of i.v – define

Line 72: Reference 13 predates the decision in 2022 – is the reference correct?

Line 78: need to describe the participating centres and how many children were treated in each centre

Line 83: Is reference 13 correct?

Line 88: “..or native vertebral infection”.

Line 94: Is the correct way of referencing Microsoft Access? Version etc?

Line 98: define IRB

Line 103: I think reference 13 is incorrect

Line 111: “…were male, …”

Table 1. Numerous grammatical issues that require addressing including the bacterial nomenclature. Why does the ID go up to 34, and why are 14,27,28,29,30 missing? Need to define MRSA/MSSA. What was the isolate from patient 35 (MRSA/MSSA), and what sample type. I count only 4 MRSA bacteraemias. Could the authors check that the details in the text matches the details in table 1 and 2. I count 28 initially treated with another antibiotic, 19 receiving a beta-lactam (including the carbapenem), 15 treated with a glycopeptide, and 6 treated with daptomycin.

Line 126. The recommended dose is 18mg/kg for children from 6 to 18 years, and 22.5mg/kg from 3 months to less than 6 years.  Why did patient 6 get the larger dose?

Line 129. 51 or 61 doses?

Line 130; 15 patients received more than one dose, weekly in 5 cases or fortnightly in nine patients (5+9=14)?

Table 2: Numerous grammatical issues that require addressing

The frustrating thing about reading the paper is having to flick between table 1 and 2. I recognise that the authors are trying to represent a lot of data, but some of the columns in table 2 could be removed, and the details included in the text (e.g. dosage data and the days to discharge). Could the authors consider redrafting the table so that the salient points are included in one figure?

Line 140: 61 or 51?

Line 166: Sentence uses “study” twice

Line 168: define AE

Line 201: revise “more charming”

Line 217: “patients”

Comments on the Quality of English Language

See above

Author Response

Reviewer 1

I should commend the authors for producing the manuscript, but there are a few minor issues to address. I’d consider revising the title – perhaps “the real-world experience of…”

Re. We thank the reviewer for his favorable evaluation and for the many hints aimed at improving our manuscript. The title was modified according to his/her suggestion.

  • Line 24: Revise “as in-patient”
    • : this was done
  • Line 29: Median
    • : this was done
  • Line 30  and through-out the manuscript – beta-lactam
    • : this was done
  • Line 30-31: The numbers in the abstract seem to be different to the data provided in the tables.
    • : Apologies for the mistake. This was modified
  • Line 32: is it 51 or 61 doses?
    • : 61. Apologies for the mistake. This was modified
  • Line 48: Conventionally written as met(h)icillin
    • : this was done
  • Line 62: First use of i.v – define
    • : this was done
  • Line 72: Reference 13 predates the decision in 2022 – is the reference correct?
    • : Apologies for the mistake. This was modified into 12
  • Line 78: need to describe the participating centres and how many children were treated in each centre
    • : this was done
  • Line 83: Is reference 13 correct?
    • : Yes.
  • Line 88: “..or native vertebral infection”.
    • : this was done
  • Line 94: Is the correct way of referencing Microsoft Access? Version etc?
    • : this was specified.
  • Line 98: define IRB
    • : this was done
  • Line 103: I think reference 13 is incorrect
    • : Indeed, sorry for this mistake. This ref was modified together with the numbers in the list, accordingly.
  • Line 111: “…were male, …”
    • : this was done
  • Table 1. Numerous grammatical issues that require addressing including the bacterial nomenclature. Why does the ID go up to 34, and why are 14,27,28,29,30 missing? Need to define MRSA/MSSA. What was the isolate from patient 35 (MRSA/MSSA), and what sample type. I count only 4 MRSA bacteraemias. Could the authors check that the details in the text matches the details in table 1 and 2. I count 28 initially treated with another antibiotic, 19 receiving a beta-lactam (including the carbapenem), 15 treated with a glycopeptide, and 6 treated with daptomycin.
    • : some patients initially reported to the data-base were then excluded because of insufficient information. The IDs were not modified. The numbers of patients treated with different antibiotics have been aligned between the table and the text. Case 35 isolate was defined (swab) and PVL defined in legend.
  • Line 126. The recommended dose is 18mg/kg for children from 6 to 18 years, and 22.5mg/kg from 3 months to less than 6 years.  Why did patient 6 get the larger dose?
    • : no specific medical reason is reported.
  • Line 129. 51 or 61 doses?
    • : 61. Apologies for the mistake. This was modified
  • Line 130; 15 patients received more than one dose, weekly in 5 cases or fortnightly in nine patients (5+9=14)?
    • : detail on the remaining patient were given in the text.
  • Table 2: Numerous grammatical issues that require addressing
    • : we revised the entire table
  • The frustrating thing about reading the paper is having to flick between table 1 and 2. I recognise that the authors are trying to represent a lot of data, but some of the columns in table 2 could be removed, and the details included in the text (e.g. dosage data and the days to discharge). Could the authors consider redrafting the table so that the salient points are included in one figure?
    • : In order to make life easier, we have incorporated both tables in one.
  • Line 140: 61 or 51?
    • : 61. Apologies for the mistake. This was modified
  • Line 166: Sentence uses “study” twice
    • : this was done
  • Line 168: define AE
    • : this was done
  • Line 201: revise “more charming”
    • : this was modified
  • Line 217: “patients”
    • : this was done

Reviewer 2 Report

Comments and Suggestions for Authors

I have reviewed the manuscript "Dalbavancin for treatment of soft tissue and bone infection in children in real life: safe, effective and hospital time sparing" submitted for consideration for publication in Children.

I have the following comments:

General comments:

·      The study investigates an important clinical question regarding the efficacy and safety of dalbavancin for treating gram-positive severe infections in children. Real-world evidence is valuable to inform practice.

·      The methods are appropriate for an observational, retrospective analysis. The number of subjects is reasonably sized, given the specific patient population. Multiple centres were included to capture a range of experiences.

·      Overall, the paper is well-written, clear, and concise. The tables effectively summarise vital data. The discussion and conclusions are balanced based on the results.

Specific comments:

·      In the Introduction, cite rates of morbidity, mortality, hospitalisations, and costs associated with Gram-positive infections in children precisely, rather than just citing the European adult data. This helps frame the significance.

·      Consider analysing length of stay before vs after the first dose of dalbavancin as an additional measure of its impact, rather than just reporting the total length of stay.

·      Elaborate on the determination and significance of the median 17-day delay before starting dalbavancin. Why was there a delay? Does it impact the assessments of safety, efficacy, and discharge timing?

·      Discuss any failures of dalbavancin therapy requiring a change to other antibiotics. If none existed, state so.

·      Address generalisability to other institutions and geographies. Consider adding limitations regarding sample size and single country.

Major revisions:

·      Provide more specifics on the clinical decision-making process for switching antibiotics to dalbavancin, including details on why there was a median 17-day delay. This context helps inform the real-world application.

·      Compare the length of hospital/antibiotic treatment pre- and post-dalbavancin to quantify its impact more directly on expediting discharge.

·      Discuss any failures of dalbavancin requiring alternative antibiotics and how often this occurred. Clearly state if there were none.

Minor revisions:

·      Cite paediatric-specific rates for morbidity, costs, etc., associated with Gram-positive infections rather than only adult statistics.

·      Consider analysing length of stay before vs after the first dose of dalbavancin as an additional metric.

·      In the Discussion, comment on generalizability to other institutions and geographical limitations.

·      Carefully proofread the manuscript to fix minor grammar, spelling, or formatting issues.

In summary,  this manuscript needs minor revisions noted above. Additional details will enhance completeness and scientific soundness.

Please do not hesitate to contact me if you have any questions. I look forward to reviewing the revised manuscript.

Comments on the Quality of English Language

Only minor editing is required.

Author Response

Reviewer 2

 General comments:

  • The study investigates an important clinical question regarding the efficacy and safety of dalbavancin for treating gram-positive severe infections in children. Real-world evidence is valuable to inform practice.
  • The methods are appropriate for an observational, retrospective analysis. The number of subjects is reasonably sized, given the specific patient population. Multiple centres were included to capture a range of experiences.
  • Overall, the paper is well-written, clear, and concise. The tables effectively summarise vital data. The discussion and conclusions are balanced based on the results.

Specific comments:

  • In the Introduction, cite rates of morbidity, mortality, hospitalisations, and costs associated with Gram-positive infections in children precisely, rather than just citing the European adult data. This helps frame the significance.

Re.: This was done.

  • Consider analysing length of stay before vs after the first dose of dalbavancin as an additional measure of its impact, rather than just reporting the total length of stay.

Re.: This was done.

  • Elaborate on the determination and significance of the median 17-day delay before starting dalbavancin. Why was there a delay? Does it impact the assessments of safety, efficacy, and discharge timing?

Re.: This was done.

  • Discuss any failures of dalbavancin therapy requiring a change to other antibiotics. If none existed, state so.

Re.: This was done.

  • Address generalisability to other institutions and geographies. Consider adding limitations regarding sample size and single country.

Re.: This was done.

Major revisions:

  • Provide more specifics on the clinical decision-making process for switching antibiotics to dalbavancin, including details on why there was a median 17-day delay. This context helps inform the real-world application.

            Re.: The following statement was introduced in the discussion: “Most patients had initial therapy with a beta-lactam agent, and then shifted to a glycopeptide agent (usually vancomycin), also due to evidence of isolation of a gram-positive microorganism. The choice to introduce dalbavancin was usually driven by the wish of consolidating clinical response, achieved by vancomycin (adopted in 80% of the patients, while shortening the hospital stay. All the above may explain the 17 days median duration of hospital stay before administration of dalbavancin.”

  • Compare the length of hospital/antibiotic treatment pre- and post-dalbavancin to quantify its impact more directly on expediting discharge.

     Re.: The following statement was introduced in the discussion: “This is well reflected in the comparison of the extended duration of hospital stay before administration of dalbavancin (median, 21 days) with the number of days between dalbavancin administration and discharge, which in 20 patients ranged between <1 and 3 days (median, 1 day), with six additional patients receiving dalbavancin first dose after discharge, as outpatient care.”

  • Discuss any failures of dalbavancin requiring alternative antibiotics and how often this occurred. Clearly state if there were none.

Re.: The following statement was introduced in the discussion: “We did not observe any treatment failure of dalbavancin therapy. None of the 31 patients reported need for alternative/additional antibiotic therapy, nor any disease recurrence or any adverse effect during a follow-up time ranging between 1 and 14 months (median, 2 months).”

Minor revisions:

  • Cite paediatric-specific rates for morbidity, costs, etc., associated with Gram-positive infections rather than only adult statistics.

Re.: We have introduced in the introduction a paragraph reporting data on children from the US (Refs 2-4). 

  • Consider analysing length of stay before vs after the first dose of dalbavancin as an additional metric.

            Re.: This was done (see above).

  • In the Discussion, comment on generalizability to other institutions and geographical limitations.

Re.: The following statement was introduced in the discussion: “The participating centers are scattered all-over the country (one in the north, one in the center, and three in the south), and range from large, pediatric referral hospitals to smaller pediatric wards embedded in large general hospitals. Thus, we do not see limitation deriving from geographic or setting type reasons to wider use of dalbavancin in such patients.”

  • Carefully proofread the manuscript to fix minor grammar, spelling, or formatting issues.

Re.: This was done.

Round 2

Reviewer 2 Report

Comments and Suggestions for Authors

For the revised manuscript, there are still a few comments I would like to add and suggest the authors make changes accordingly:

·      Provide more details on the specific inclusion/exclusion criteria used to select patients for analysis. Were all possible patients treated with dalbavancin included, or was a subset selected? This can clarify any selection bias.

·      Discuss the limitations of the retrospective design and small sample size from only 5 centres in the Discussion.

In conclusion, I think this study provides valuable real-world evidence on the utility of dalbavancin for treating ABSSSIs and osteomyelitis in children. Addressing the comments above will further improve the quality of the work. I recommend accepting this manuscript for publication pending minor revisions.

Comments on the Quality of English Language

Minor editing is still required. Please check for spelling errors and typos.

Author Response

Thank you for your comments, we have revised them.